# Multilocation Evaluation of Virginia and Runner -Type Peanut Cultivars for Yield and Grade in Virginia–Carolina Region

**Naveen Kumar [1], David C. Haak [2], Jeffrey C. Dunne [3] and Maria Balota [1,\*]**

[1] Tidewater Agricultural Research and Extension Center, Virginia Tech, Suffolk, VA 23437, USA
[2] School of Plant and Environmental Sciences, Virginia Tech, Blacksburg, VA 24060, USA
[3] Department of Crop and Soil Sciences, North Carolina State University, Raleigh, NC 27695, USA
[\*] Correspondence: mbalota@vt.edu

**Abstract:** The peanut is mostly grown in semi-arid tropical regions of the world, characterized by unpredictable rainfall amounts and distribution. Average annual precipitation in the Virginia–Carolina (VC) region is around 1300 mm; however, unpredictable distribution can result in significant periods of water deficit and subsequent reduction in yield and gross income. The development of new peanut cultivars with high yield and acceptable levels of yield stability across various water-availability scenarios is an important component of the peanut breeding program in Virginia and the Carolinas, where the large-seeded Virginia-type peanut is the predominantly grown market type. In addition, the simultaneous use of runner cultivars developed in the dryer southeastern region has been proposed as a practical solution to limited irrigation availability in the VC region. Still, the identification and adequate utilization of available commercial cultivars with the best combination of yield, drought tolerance, and gross income is more immediately beneficial to the peanut industry, yet this assessment has not been carried out to date. The aim of this study was to identify cultivars that maintain high yield and grade, therefore gross income, across a wide range of environmental conditions. We evaluated five commercially available Virginia and runner-type peanut cultivars for pod yield stability using multilocation trials over four years across 13 environments. Additive main effects and multiplicative interaction (AMMI) and different stability approaches were used to study genotype (G), environment (E), and their interaction (G × E) on pod yield. Pod yield stability was specifically assessed by using the Lin and Binn approach, Wricke's ecovalence, Shukla's stability, and the Finlay–Wilkinson approach. The combined analysis of variance showed highly significant effects ($p \leq 0.001$) for genotypes, environments, and G × E for pod yield. The environments varied in yield (2840–8020 kg/ha). Bailey, Sullivan, and Wynne are Virginia-type cultivars. The grade factors SMK, SS, and TK changed with water regime within both market types. Among the runner cultivars, TUFRunner 297 presented high mean productivity; however, it showed specific adaptation to limited environmental conditions. Based on different stability approaches, this study concludes that Sullivan and Bailey are the most stable and adaptable cultivars across the testing environments, whereas Wynne exhibited specific adaptability to some environments. These findings have important implications for peanut cultivar recommendations in terms of meeting peanut industry standards for yield, grading quality, and breeding progress.

**Keywords:** peanut; genotype by environment interaction; stability analysis; adaptability; pod yield

## 1. Introduction

The US produced 1.5 billion USD worth of peanut (*A. hypogaea* L.) in 2021, of which 15% was contributed by the Virginia–Carolina (VC) region [1]. Peanut production in the VC region is dominated by the globally recognized, large-seeded Virginia-type peanut in the United States. In contrast, a runner-type peanut is primarily grown in the lower southeastern production areas of the US. The Virginia-type peanut is the predominant market type grown in the VC region because of the desirability of its large pod and

kernel size, but occasionally runners are grown. For kernel size, Virginia-type peanuts are almost twice the size of runners, for which growers can receive premiums. Importantly however, there are considerable differences in the cost of production between Virginia and runner types. The total estimated cost of production for Virginia-type peanuts requires an additional 231.30 USD per hectare in inputs compared to runner-type peanuts [2]. For Virginia-type, high yield and grade factors, including high extra-large kernel (ELK) content, is achieved at the cost of higher requirements of calcium than runner types [3]. In addition to higher calcium requirements, large-size Virginia-type peanuts require more soil moisture during planting than runner type for successful germination [4]. This can lead to increased cost from extra seeds to compensate for germination failures, especially in dry years, or supplemental costs incurred for irrigation to assist plant stand establishment. The Virginia market type requires around 135 to 155 days after planting (DAP) to reach maturity, while runners need over 155 DAP in the VC region [5]. The late maturity of runner types in the VC region results in potential for freeze damage and a subsequent yield penalty. The majority (95%) of peanut production in the VC region is rainfed, which leads to occasionally erratic episodes of water deficit. Yield and quality of peanut under rainfed conditions are significantly reduced by drought conditions, including periodic drought [6]. The duration and intensity of periodic drought vary by year and location. Drought conditions in peanut also enhance the growth of *Aspergillus flavus*, which causes aflatoxin contamination in seeds. Aflatoxin-contaminated peanuts reduce crop market value and cause diseases in humans [7]. Breeding for improved drought tolerance in peanut might be a successful approach for the reduction of aflatoxin contamination, suggested by Guo et al. [8]. Some studies have shown that peanut cultivars with drought-tolerance attributes were more resistant to aflatoxin contamination [9,10]. Peanut cultivation cost per hectare is about 2223 to 2346 USD, varying across different growing regions of the VC region [11]. Therefore, based on the cost per hectare, economic sustainability can be achieved if peanut yields are greater than 4500 kg/ha, which can be attained by minimizing the effects of biotic and abiotic factors on yield and quality [12]. The gross income of the growers is dependent on both yield and quality, thus maximizing production alone does not always equate to maximum economic returns. Therefore, successful peanut breeding programs focus on delivering cultivars with guaranteed superior yield and quality performance across a wide range of environmental conditions. These cultivars can increase productivity and growers' incomes.

The combination of drought stress and crop management account for yearly variations that leads to genotype by environment interaction (G × E), therefore affecting yield stability [13–15]. The G × E refers to differential responses of genotypes across diverse environments [16]. This is a crucial consideration when cultivars will be exposed to varied conditions in the production settings that cannot be predicted in advance. The interaction of G × E can complicate the selection of stable and ideal genotypes across environmental conditions. Numerous methods have been employed to quantify cultivars' range of performance across testing locations, including direct modeling of the G × E interaction for yield. The propensity to model the G × E led to the development of a series of methods and approaches for multi-environment trials called "stability analyses" [17]. The best genotype in one environment may not necessarily be the best genotype in another environment [18–21]. The "agronomic concept" of stability defines a stable genotype as one which minimally contributes to the G × E and how the genotype responds to the change in environment [17]. To be widely accepted, a cultivar must perform well across multiple environments before registration and release. In this scenario, interpreting G × E effects in multi-environment trials along with the application of different stability analysis models assists in the selection of stable genotypes for a wide range of environments [22]. For analysis of multilocation trials, the additive main effects, multiplicative interactions (AMMI) [23,24], and different stability models such as Lin and Binn's, Wricke's ecovalence, Shukla's stability, and Finlay–Wilkinson's have been used to study G × E effects [17,22–26]. It is expected that each stability parameter shows a different ranking pattern of the cultivars. Hence, it is best to

compute stability using all methods and, finally, to select the superior genotypes based on their combined stability. Some stability models are based on the genotypic contribution to the GE variance, and others on G + (G × E) (e.g., univariate or multivariate). The models based on the G + (G × E) are more repeatable if calculated within mega-environments, because mean yield is more repeatable. Recently, several stability parameters have been used as phenotypes in genome-wide association studies (GWAS) to identify novel genomic loci associated with the G × E interaction. Correspondence between conventional stability estimates alongside a mixed model for grain yield in soybeans showed independent results. These findings may be influenced by incomplete and unbalanced data structure used in multi-environment trials [27]. This is a common occurrence in field trials and often suggested for more genotypes and environments. To address this limitation, researchers interested in studying the adaptation through G × E need to consider the potential influences of modelling approaches on their desired outcome.

The goal of this study was to identify cultivars that maintain high yields across a wide range of environmental conditions. Specific objectives were (1) to compare the yield and grade performance of Virginia and runner-type peanut cultivars across different growing environments in the VC region, and (2) to investigate the G × E and yield stability of five Virginia and runner-type peanut cultivars across 13 environments.

## 2. Materials and Methods

### 2.1. Plant Material and Experimental Design

A total of five peanut cultivars, three Virginia-type (Bailey [28], Sullivan [29], and Wynne [29]) and two runner-type (FloRun™ '107' [30], TUFRunner™ '297' [31]) were evaluated for four years (2016, 2017, 2018, and 2020) at four locations in Virginia and North Carolina under two levels of water regime, i.e., rainfed and irrigated (rainfed plus irrigation) (Table 1).

**Table 1.** Description of Virginia and runner cultivars tested from 2016 to 2020, and the reasons for inclusion in the testing.

| Name | Genotype | Type | Use | Year /Location of Use in Test | Reason | References |
|---|---|---|---|---|---|---|
| FloRun '107' | G02 | Runner | Cultivar | 2016, Din; 2017, Cap, RM and Suf; 2020: Suf | High yield | [30] |
| TUFRunner '297' | G04 | Runner | Cultivar | 2016, Din; 2017 Cap, RM and Suf; 2018 RM and Suf; 2020: Suf | High oleic/high yield | [31] |
| Bailey | G01 | Virginia | Cultivar | 2016, Din; 2017 Cap, RM and Suf; 2018 RM and Suf; 2020: Suf | Widely grown/High yield | [28] |
| Sullivan | G03 | Virginia | Cultivar | 2016, Din; 2017 Cap, RM and Suf; 2018 RM and Suf; 2020: Suf | High oleic/TSWV resistant | [29] |
| Wynne | G05 | Virginia | Cultivar | 2016, Din; 2017 Cap, RM and Suf; 2018 RM and Suf; 2020: Suf | High oleic/large kernels | [29] |

The trials were conducted at the Tidewater Agricultural Research and Extension Center (TAREC) in Suffolk, VA (36°68′ N, 76°77′ W, 25 m elevation), the Upper Coastal Plain Research Station (UCPRS) near Rocky Mount, NC (35°57′ N, 77°48′ W, 33 m elevation), and two farmers' fields in Capron (36°42′ N, 77°12′ W, 34 m elevation) and Dinwiddie (37°04′ N, 77°37′ W, 78 m elevation), VA. Soils for the four locations were Eunola–Kenansville soil (fine–loamy, siliceous, thermic Aquic Hapludults) at Suffolk, Goldsboro; sandy loam soil (fine–loamy, siliceous, thermic Aquic Paleudalts) at Rocky Mount; Nansemond soil (fine–loamy, Aquic Hapludults) at Capron; and Helena sandy loam soil (fine, semiactive, thermic Aquic Hapludults) at Dinwiddie. A detailed description of environments, which were represented by combination of year, location, and water regime is presented in Table 2. Cultivars were planted in two-row plots of 10.6 m long × 0.9 m wide, using a randomized complete block design with four replications within each water regime. The seeding rate was 13 seeds m$^2$ with 4.57 m alley in between each replication. Cultural practices were performed based on the recommendations of the *Virginia Peanut Production Guide* [12]. Air temperature,

relative humidity (RH), and rainfall were continuously monitored next to the plots with a weather station (Watchdog Temperature/RH Station, Model 2450, Spectrum Technologies Inc., Plainfield, IL, USA). In addition, climate data for precipitation and temperature for all years were collected from PRISM climate group from the nearest weather station to the field trials. Based on peanut growth stage and weekly precipitation requirement, plots were irrigated using a lateral-pull boom cart sprinkler irrigation system (E1025 Reel Rain, Amadas Ind., Suffolk, VA, USA) shown in Figure 1. Plots were dug from mid-September to early October at harvest maturity [32] with a KMC two-row digger. Pods were combined after a few days of windrow drying with a Hobbs peanut combine (Model 325A). Yield samples were cleaned and dried down to 10% moisture. Pod yield was calculated from plot weight and adjusted to 7% seed moisture and foreign material (FM) percentage. The gross income was calculated from yield and corrected for seed moisture, foreign material, and price using the grade traits according to the USDA Agricultural Marketing Service formula [33]. The raw data for yield and other grade factors are available in Supplementary Table (Table S1).

**Table 2.** Description of 13 peanut growing environments with location, water regime, and soil type in tests from 2016 to 2020.

| Season | Environment | ENV | Location | Regime | Soil Type |
|--------|-------------|-----|----------|--------|-----------|
| 2016 | Dwd16RfIR | E04 | Dinwiddie, VA | Rainfed and Irrigated | Helena sandy loam |
|      | Dwd16Rf | E03 | Dinwiddie, VA | Rainfed | Helena sandy loam |
| 2017 | Cap17RfIR | E02 | Capron, VA | Rainfed and Irrigated | Nansemond |
|      | Cap17Rf | E01 | Capron, VA | Rainfed | Nansemond |
|      | Rkm17RfIR | E06 | Rocky Mount, NC | Rainfed and Irrigated | Goldsboro sandy loam |
|      | Rkm17Rf | E05 | Rocky Mount, NC | Rainfed | Goldsboro sandy loam |
|      | Suf17Rf | E09 | Suffolk, VA | Rainfed | Eunola–Kenansville |
| 2018 | Rkm18RfIR | E08 | Rocky Mount, NC | Rainfed and Irrigated | Goldsboro sandy loam |
|      | Rkm18Rf | E07 | Rocky Mount, NC | Rainfed | Goldsboro sandy loam |
|      | Suf18RfIR | E11 | Suffolk, VA | Rainfed and Irrigated | Eunola–Kenansville |
|      | Suf18Rf | E10 | Suffolk, VA | Rainfed | Eunola–Kenansville |
| 2020 | Suf20RfIR | E13 | Suffolk, VA | Rainfed and Irrigated | Eunola–Kenansville |
|      | Suf20Rf | E12 | Suffolk, VA | Rainfed | Eunola–Kenansville |

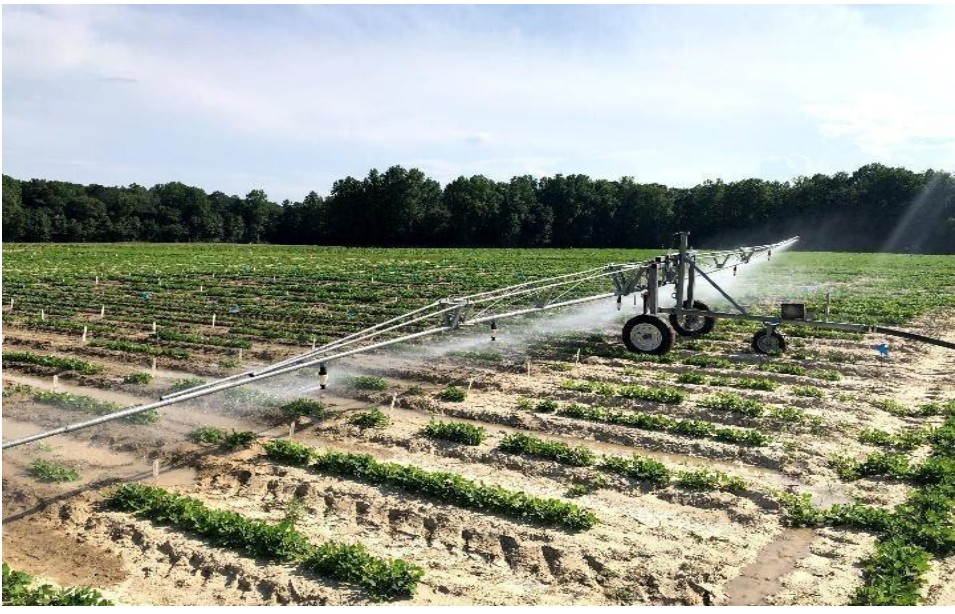

**Figure 1.** Lateral-pull boom cart sprinkler irrigation system used to irrigate the plots.

Grade characteristics measured included ELK, kernels not passing a 25.4 mm (1 in) × 8.5 mm (21.5/64 in) screen, sound mature kernels (SMK), and undamaged kernels not passing a 25.4 × 6 mm screen for Virginia types or a 19.05 × 6.4 mm slotted-screen for runner types [34].

The damaged kernels (DK) percentage represented decayed, molded, sprouted, or discolored kernels due to insects or weather damage; sound splits (SS) were undamaged halved or broken kernels with no penalty up to 4% [34] and the other kernels (OK) were those passing through a 25.4 × 5.9 mm (15/64 in) screen determined by subjecting a 500 g pod sample to the Federal state inspection procedure. The sum of SMK, DK, SS, and OK constituted the total kernels (TK). The gross value was determined from yield and grade standards from the federal formula [34]. Yield and grade data from individual trials were collected in all environments.

### 2.2. Statistical Analysis

Yield and grade data was subjected to analysis of variance (ANOVA) using a fitted mixed model to assess the magnitude of G × E, where genotype (G) was considered as a fixed effect, and growing environment (E) as a random effect. Boxplots for grading factors (SMK, SS, and TK) were created to explore the heterogeneity of genetic variance using the "statgenGXE" package [35]. The means for each grading factor and market type were compared with independent *t*-tests and a significance level of 0.05 was used to judge the significance. There is no one biometrical model that can adequately explain the stability performance of genotypes across environments [19]. Therefore, different statistical approaches and models were used to avoid the limitations of any single model. In this study, analysis of yield stability was done using Wricke's ecovalence (Wi), Shukla's stability variance ($\sigma^2$i), Finlay and Wilkinson's joint regression analysis, Lin and Binn's superiority measures (Pi), and AMMI. Wricke's ecovalence stability coefficient shows the contribution of each genotype to the G × E, squared and summed across environments, in an unweighted analysis of the G × E means. The formula for this model is:

$$Wi = \Sigma \ (Y_{ij} - Y.j - Yi + Y...);$$

where $Y_{ij}$ = mean of genotype i in environment j, Y. j = mean yield of genotype across environments, Yi = environment mean, Y . . . = Overall mean [25,36]. Wricke ecovalence evaluates the yield dynamic stability of the cultivar which is desirable for breeding in water-limited conditions with inconsistent rainfall patterns. In Wricke's ecovalence, cultivars with the lowest values have smaller deviations from the mean across environments, and contributed least to the G × E interaction and are, therefore, more stable. Shukla's stability variance ($\sigma^2$i) is defined as the variance around the genotype's phenotypic mean across all environments [24]. It measures each genotype's contribution to the overall G × E and error term. The superiority measure was proposed by Lin and Binns [25]. The cultivar superiority measure is a function of the sum of the squared differences between a cultivar's mean and the best cultivar's mean, where the sum is across environments. The formula for this model is as follows:

$$Pi = \Sigma \ (X_{ij} - M_j)2/(2n);$$

where Pi is the superiority measure of the $i^{th}$ cultivar, $X_{ij}$ is the yield of the $i^{th}$ cultivar grown in the $j^{th}$ environment, $M_j$ is the maximum yield value among all cultivars in the $j^{th}$ environment, and *n* is the number of locations. Finlay and Wilkinson's joint regression coefficient is the response of genotype to the environmental index that is derived from the average performance of all genotypes in each environment [26]. The model fitted in Finlay–Wilkinson analysis is

$$Y_{ij} = \mu + Gi + \beta iEj + \varepsilon ij;$$

where Y*ij* j is phenotypic value of genotype *I* in the environment *j*, μ is the general mean, G*i* is genotypic effect, β*i* is sensitivity parameters, E*j* is environment effect, and εij is residual. According to Finlay and Wilkinson, regression coefficients approximating to 1.0 indicate average stability; however, it must always be associated and interpreted with the mean yield to determine adaptability. An AMMI stability model is one of the most widely used tools in multiple-environment trials to understand complex G × E and increase the accuracy

to improve recommendations, selection, and genetic gains [36]. AMMI first performs an analysis of variance (AOV) to partition the variance into G, E, and G × E interaction effects, and then it applies principal components analysis (PCA) to G × E. Here, an AMMI model was used to estimate the response variable for the *i*th cultivar in the *j*th environment, given as follows:

$$Y_{ij} = \mu + \alpha i + \tau j + \Sigma_{k=1}^{p} \lambda k \alpha i k t j k + \varepsilon i j$$

where $Y_{ij}$ is the yield of the $i^{th}$ genotype in the $j^{th}$ environment; $\mu$ is the grand mean; $\alpha i$ and $\tau j$ are the genotype and environment deviations from grand mean, respectively; $\lambda k$ is the eigenvalue of the PCA analysis axis; $\varepsilon i j$ is the residual.

The AMMI biplot graphs show the dispersion of genotypes, environments, and interactions between them. Biplots in AMMI1 identify genotypes that are adapted to a particular environment or broadly adapted. The biplot origin represents the overall phenotypic mean and yield. The position of genotype and environment from the origin provides insight into the G × E interaction. Genotypes near the origin are insensitive to environmental interaction, hence are broadly adapted. All statistical analyses involving G × E, AMMI model, and stability indices were performed in R using the packages "statgenGXE" [35] and "metan" [37]. Figures were produced using the package ggplot2 [38].

## 3. Results and Discussion

The environments included in the present study differed in soil type and weather conditions, especially rainfall. The amount and distribution of rainfall varied during the growing seasons at the Suffolk location in 2017, 2018, and 2020. Rainfall was uniform throughout the growing season in 2017, but in 2018 it was reduced in the latter part of the season, whereas in 2020 the month of July had a rainfall deficit, which affected peg and pod development by prolonging the vegetative phase of peanuts (Table 3). For example, the 53 mm rainfall in 2020 was received in the last week of July, while during the first 3 weeks and the last part of June no precipitation was recorded. On sandy soils, this prolonged absence of precipitation made year 2020 one of the driest, and unsuitable for peanut production in Virginia. As a result, in 2020, the Virginia State peanut production was 560 kg/ha less than in 2019, a year with good precipitation, which agrees with our data [39]. At Rocky Mount, the minimum (MIN) and maximum (MAX) temperatures in 2018 were higher than in 2017, but all environments met the sufficient heat unit requirement for peanut maturity and growth [40]. (Table 3). Peanut trade considers both yield and grade characteristics to determine peanut price [33]. For this reason, growers take their crop to buying points where samples from each truck load are instantly taken for grading to be performed by specialized grading services under the Department of Agriculture and Consumer Services in each state. For some grade characteristics, such as ELK, SMK, and TK, growers receive price premiums, while for others, such as DK, deductions. Therefore, the gross income a grower may receive is based on a federal formula accounting for all grades determined, as in this work, in addition to yield. The increase in average temperature was predicted to decrease the gross income from peanuts [41]. Indeed, a slight decrease in gross income was recorded in 2018 compared to 2017 at Rocky Mount (Figure 2). Gross income was greater under the irrigated regime compared to rainfed. Virginia type (2191 ± 378 USD/ha) cultivars provided higher gross returns then runner types (2079 ± 407; $p = 0.02$) (Figure 2) in the study.

**Table 3.** Weather information from seven environments (locations and years) recorded during the peanut growing season. $GDD_{13.3}$ is the growing degree days using a base temperature of 13.3 °C.

| Location | Dinwiddie (VA) | Capron (VA) | Rocky Mount (NC) | | Suffolk (VA) | | |
|---|---|---|---|---|---|---|---|
| **Year** | **2016** | **2017** | **2017** | **2018** | **2017** | **2018** | **2020** |
| | | | **MAX temp (°C)** | | | | |
| May | 23 | 25 | 26 | 28 | 26 | 28 | 25 |
| June | 29 | 29 | 30 | 31 | 32 | 31 | 29 |
| July | 32 | 32 | 32 | 31 | 33 | 31 | 34 |
| Aug | 32 | 29 | 30 | 31 | 31 | 32 | 32 |
| Sep | 28 | 27 | 28 | 31 | 28 | 31 | 26 |
| Oct | 23 | 25 | 25 | 30 | 27 | 25 | 24 |
| | | | **MIN temp (°C)** | | | | |
| May | 13 | 14 | 15 | 18 | 14 | 18 | 16 |
| June | 17 | 18 | 19 | 21 | 21 | 19 | 19 |
| July | 21 | 21 | 22 | 21 | 22 | 20 | 22 |
| Aug | 20 | 20 | 21 | 21 | 19 | 21 | 22 |
| Sep | 19 | 16 | 17 | 21 | 16 | 20 | 17 |
| Oct | 14 | 14 | 12 | 18 | 16 | 13 | 11 |
| | | | **Monthly cumulative GDD13.3 (°C)** | | | | |
| May | 141 | 188 | 219 | 291 | 195 | 285 | 144 |
| June | 298 | 308 | 321 | 366 | 318 | 357 | 300 |
| July | 401 | 414 | 425 | 388 | 415 | 371 | 442 |
| Aug | 401 | 342 | 365 | 393 | 354 | 402 | 417 |
| Sep | 303 | 251 | 268 | 368 | 268 | 363 | 258 |
| Oct | 51 | 65 | 70 | 108 | 74 | 106 | 39 |
| | | | **Rainfall (mm)** | | | | |
| May | 219 | 137 | 125 | 165 | 119 | 104 | 74 |
| June | 128 | 105 | 122 | 74 | 89 | 104 | 97 |
| July | 123 | 133 | 150 | 180 | 61 | 208 | 53 |
| Aug | 51 | 204 | 175 | 94 | 185 | 150 | 216 |
| Sep | 159 | 40 | 76 | 130 | 94 | 112 | 259 |
| Oct | 120 | 6 | 86 | 3 | 58 | 58 | 20 |

The distribution of five Virginia- and runner-peanut cultivar means for gross income, yield, SMK, SS, and TK in 13 environments for water regime and market type are summarized through boxplots in Figures 2–6. The cultivars' yield for each trial (Figure 3) showed that irrigated plots produced more yield than rainfed plots. This agrees with findings by others [42–45]. However, based on this data, a rainfed environment produced the highest mean yield, as described later. Since in Virginia and the VC region, irrigation is not a decisive factor for high yield, only a few growers have irrigation capability. This is because the climate is sub-humid with a high relative humidity of the air during the summer months, which may make excessive rain/irrigation or humidity as detrimental as dry weather. Under a humid environment, foliar and soil borne diseases may become difficult to control and, under high disease pressure, yield is reduced. Therefore, in some years, irrigation is not needed for high yields, such as in 2017 in Suffolk, known a "good year" for peanut production in the VC region [46]. Overall, there was no significant difference in pod yield between Virginia type (5619 ± 823 kg/ha) and runner type (5426 ± 921 kg/ha; $p$ = 0.07) cultivars. The Virginia type performed better than the runner type specifically in the environments Rkm17FR (E05), Rkm17RfIR (E06), and Suf20RfIR (E13), as shown on Figure 3.

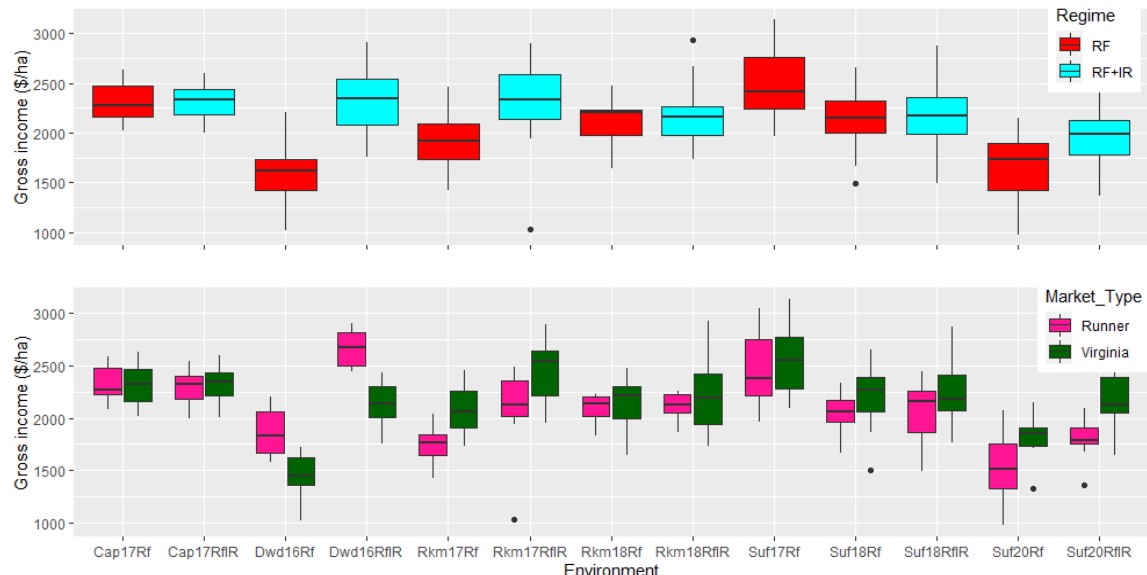

**Figure 2.** Boxplot for the effect of water regime and market type on gross income (USD/ha) of 5 Virginia and runner peanut cultivars in 13 environments. Each plot indicates total range, interquartile range (box), and median (line). Environment names are coded and the first three letters indicate location: Cap, Capron, VA; Dwd, Dinwiddie, VA; RM, Rocky Mount, NC; Suf, Suffolk, VA; two digits indicate year of the trial; 16 for 2016, 17 for 2017, 18 for 2018, 20 for 2020; and the last letters Rf and RfIR for water regime: Rf, rainfed, RfIR, rainfed plus irrigated. The upper part of the figure shows the effect of water regime and the lower part the effect of market type on the gross income in 13 environments.

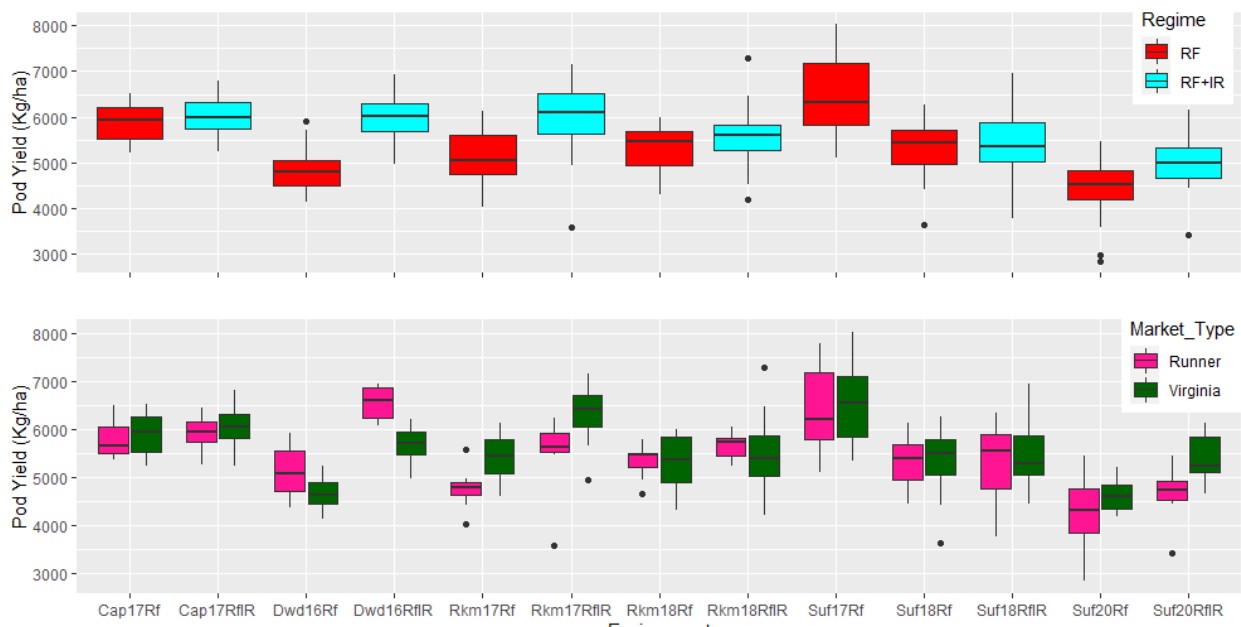

**Figure 3.** Boxplot for the effect of water regime and market type on yield of 5 Virginia and runner peanut cultivars in 13 environments. Each plot indicates total range, interquartile range (box), and median (line). Environment names are coded and the first three letters indicate location: Cap, Capron, VA; Dwd, Dinwiddie, VA; RM, Rocky Mount, NC; Suf, Suffolk, VA; two digits indicate year of the trial; 16 for 2016, 17 for 2017, 18 for 2018, 20 for 2020; and the last letters Rf and RfIR for water regime: Rf, rainfed, RfIR, rainfed plus irrigated. The upper part of the figure shows the effect of water regime and the lower part the effect of market type.

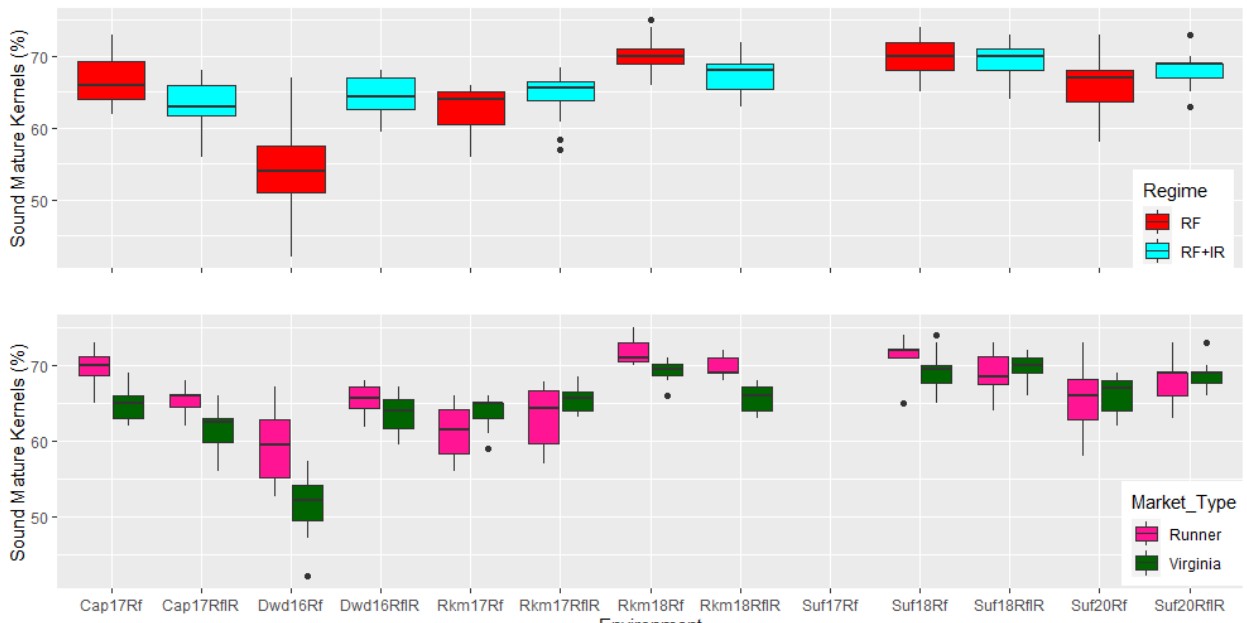

**Figure 4.** Boxplot for the effect of water regime and market type on sound mature kernels (SMK) of 5 Virginia and runner peanut cultivars in 13 environments. Each plot indicates total range, interquartile range (box), and median (line). Environment names are coded and the first three letters indicate location; Cap, Capron, VA; Dwd, Dinwiddie, VA; RM, Rocky Mount, NC; Suf, Suffolk, VA; two digits indicate year of the trial; 16 for 2016, 17 for 2017, 18 for 2018, 20 for 2020; and the last letters Rf and RfIR for water regime: Rf, rainfed, RfIR, rainfed plus irrigated. The upper part of the figure shows the effect of water regime and the lower part the effect of market type on the sound mature kernels content.

However, in 2020 alone, when comparing rainfed with rainfed plus irrigated plots, the least affected by the irrigation regime were the runners. TUFRunner297 and FloRun 107 only showed a yield reduction of 124 and 339 kg/ha, respectively, in absence of irrigation, in comparison with a yield decrease of 657 kg/ha for Bailey, 662 kg/ha for Wynne, and 1059 kg/ha for Sullivan. This agrees with classical data from Erickson and Ketring [47], showing that the the large-seeded Virginia-type requires more water to fill the pods than the runner type. The grade factors SMK, SS, and TK were also changed by the water regime within both market types in 2020. For example, SMK and TK were similarly reduced for both runners and Bailey, a relatively small seeded Virginia-type, but substantially more reduction was observed for Sullivan and Wynne; SS was slightly increased by irrigation for both market types. Nonetheless, the potential economic loss due to absence or irrigation in a dry year such as 2020 is more likely for Virginia than for runner types, i.e., gross income difference between irrigated and non-irrigated plots were 112 and 220 USD/ha for FloRun 107 and TUFRunner297, respectively, and 201, 326, and 568 USD/ha for Wynne, Bailey, and Sullivan, respectively.

The sound mature kernels (SMK) percent for runner type (66.6 ± 4.82) was significantly higher than for Virginia type (65.0 ± 5.31, $p = 0.02$) (Figure 4). A similar trend was noticed for SS, which was higher for runner type (3.53 ± 2.1) than for Virginia type (2.80 ± 1.64, $p = 0.004$) (Figure 5). Sound split content was linearly proportional with the SMK content for the runners, in agreement with the findings of Anco et al. [45]. Virginia-type cultivars exhibited lower TK than runner cultivars with 71 ± 3.07 and 75 ± 2.66 ($p = 0.0001$), respectively.

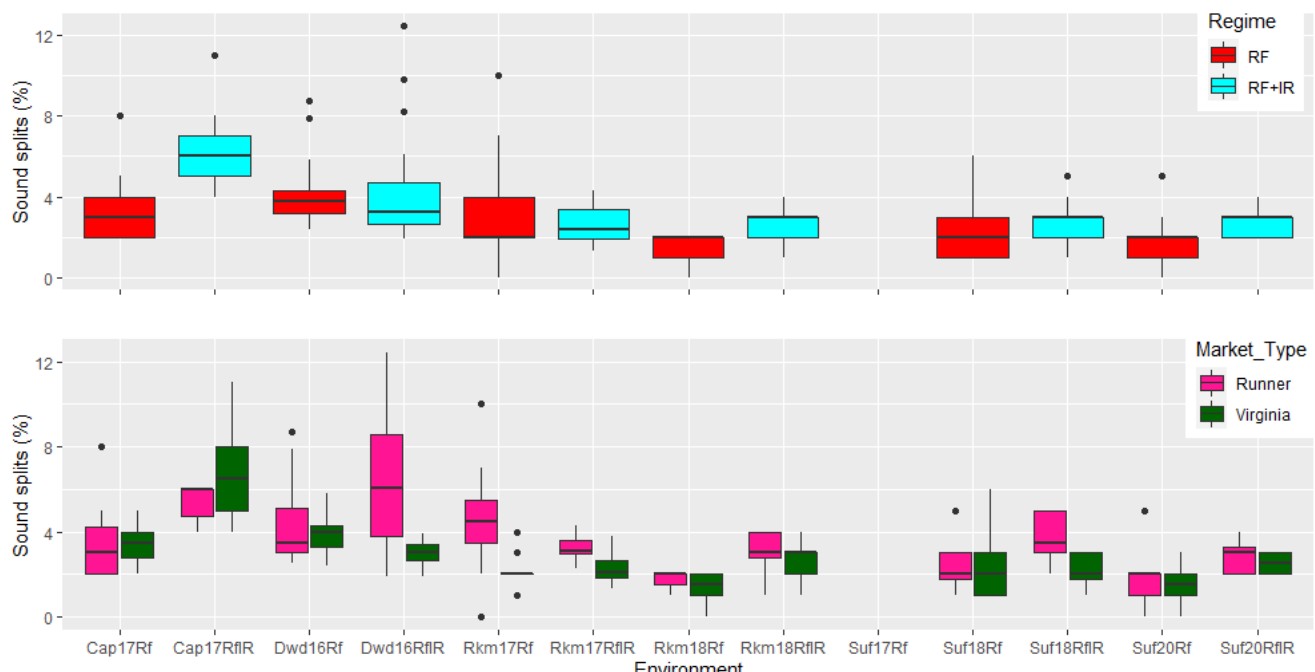

**Figure 5.** Boxplot for the effect of water regime and market type on sound splits (SS) of 5 Virginia and runner peanut cultivars in 13 environments. Each plot indicates total range, interquartile range (box), and median (line). Environment names are coded and the first three letters indicate location: Cap, Capron, VA; Dwd, Dinwiddie, VA; RM, Rocky Mount, NC; Suf, Suffolk, VA; two digits indicate year of the trial; 16 for 2016, 17 for 2017, 18 for 2018, 20 for 2020; and the last letters Rf and RfIR for water regime: Rf, rainfed, RfIR, rainfed plus irrigated. The upper part of the figure shows the effect of water regime and the lower part the effect of market type on the sound splits (SS) content.

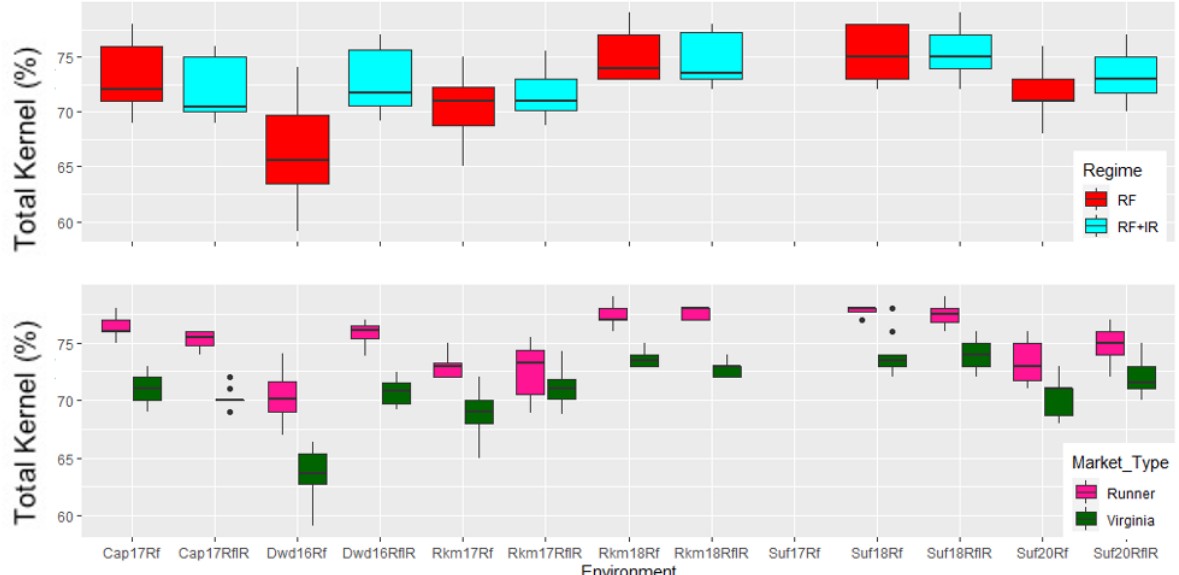

**Figure 6.** Boxplot for the effect of water regime and market type on total kernels (TK) of 5 Virginia and runner peanut cultivars in 13 environments. Each plot indicates total range, interquartile range (box), and median (line). Environment names are coded and the first three letters indicate location: Cap, Capron, VA; Dwd, Dinwiddie, VA; RM, Rocky Mount, NC; Suf, Suffolk, VA; two digits indicate year of the trial; 16 for 2016, 17 for 2017, 18 for 2018, 20 for 2020; and the last letters Rf and RfIR for water regime: Rf, rainfed, RfIR, rainfed plus irrigated. The upper part of the figure shows the effect of water regime and the lower part the effect of market type on the total kernels (TK) content.

A combined analysis of variance using a fitted mixed model was performed to assess the magnitude of G × E. The results of the combined ANOVA across 13 environments for the five Virginia and runner peanut cultivars showed that G, E, and G × E exhibited significant ($p < 0.001$) effect for pod yield (Table 4). A significant G × E indicates the need for stability analysis to evaluate the impact of each genotype on G × E, which may shape differences among genotypes in each environment.

**Table 4.** Combined analysis of variance for pod yield in the study of 5 Virginia and runner peanut cultivars in 13 environments.

| Source | DF | SS | MS | F-Value | *p*-Value |
|---|---|---|---|---|---|
| Environment (E) | 12 | 95,555,434 | 7,962,953 | 26.57969 | *** |
| Rep (E) | 39 | 19,263,173 | 493,927.5 | 1.64869 | 0.015 |
| Genotype (G) | 4 | 16,411,368 | 4,102,842 | 13.69496 | *** |
| G × E | 48 | 29,139,325 | 607,069.3 | 2.026348 | *** |
| Residuals | 182 | 54,524,984 | 299,587.8 | | |
| CV (%) | 9.88 | | | | |

DF, degree of freedom; G × E, genotype by environment interaction; SS, sum of square; MS, mean square; P, level of probability. Significant at $p \leq 0.001$ (***).

Based on the yield Wi rank, Sullivan can be regarded as the most stable followed by FloRun 107 and Bailey (Table 5). FloRun 107 ranked last in yield (Table 5). Bailey ranked top in pod yield (5796 kg/ha) and third rank in Wi stability (Table 5). Shukla provides a measure of the consistency of the genotype in a static manner i.e., static stability.

**Table 5.** Mean yield and average stability ranking of 5 Virginia- and runner-peanut cultivars tested across 13 environments.

| Genotype | GEN | Yield (kg/ha) | Lin & Binn's | Shukla's | Wricke's | FW | Average Rank |
|---|---|---|---|---|---|---|---|
| Sullivan | G03 | 5511 (3) * | (3) | (1) | (1) | (1) | 1.5 |
| Bailey | G01 | 5796 (1) | (1) | (2) | (3) | (2) | 2.0 |
| TUFRunner297 | G04 | 5622 (2) | (2) | (3) | (4) | (3) | 3.0 |
| FloRun107 | G02 | 5262 (5) | (4) | (4) | (2) | (4) | 3.5 |
| Wynne | G05 | 5344 (4) | (5) | (5) | (5) | (5) | 5.0 |

* Numbers between bracket denote rank.

The stability variance is a linear combination of the ecovalence, and therefore both Wi and $\sigma^2 i$ are equivalent for ranking purposes [48]. Shukla's stability ranking as well as the mean yield are given in Table 5. The most stable cultivars indicated by this stability measure were Sullivan and Bailey whereas Wynne was least stable across all environments. Runner type cultivar TUFRunner 297 ranked intermediately for stability with the third rank in Shukla's stability ranking. Genotypes with the smallest values of Lin and Binn's superiority tend to be more stable and closer to the best genotype in each environment. From this analysis, the most stable cultivar according to Lin and Binn's superiority was Bailey (5796 kg/ha), followed by TUFRunner 297 (5622 kg/ha), and Sullivan (5511 kg/ha). The least stable according to Lin and Binn's superiority was Wynne (Table 5).

Finlay–Wilkinson (FW) ranking analysis (Table 5) indicated that Sullivan was most stable among all cultivars and adapted to most of the environments, followed by Bailey, which had the highest mean yield. Wynne was the least stable according to FW and adapted to only a few specific environments. Figure 7 reveals a similar pattern, showing that Sullivan and Bailey are more stable. Sullivan performed well in low yielding but not in high yielding environment, whereas Bailey performed well regardless of the testing conditions (Figure 7). TUFRunner 297 produced an acceptable yield in low but very high yield in high yielding environments. Wynne yielded low in most of the environments and was less stable, but performed better in Rkm17RfIR (E06), Suf 17Rf (E09), and Cap17RfIR (E02), reflecting its narrow adaptation to the environment. The other cultivars with yields between Bailey and

Wynne in Figure 7 showed average stability according to Finlay and Wilkinson's model. The mean squares from AMMI analysis of variance (Table 6) showed significant variation among the genotypes, environments, and their interaction for pod yield.

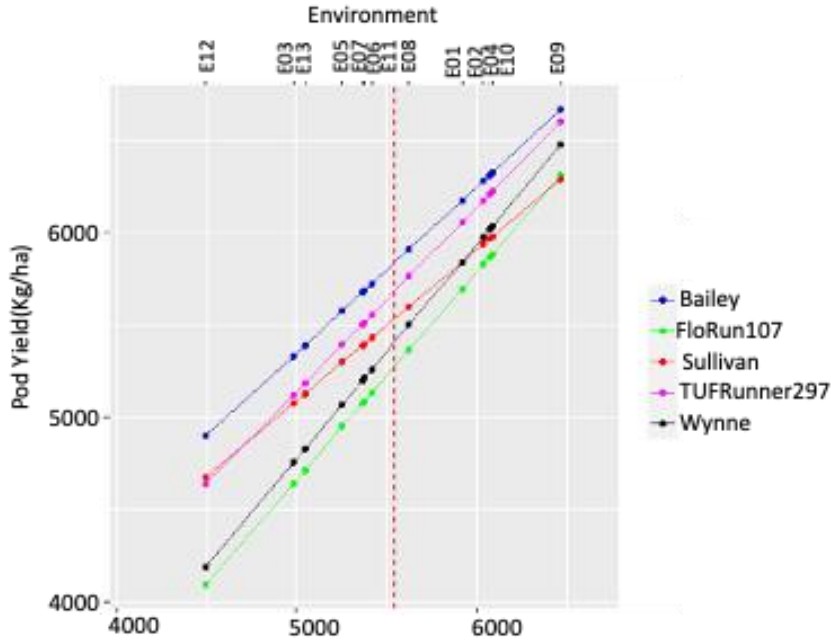

**Figure 7.** Plot of the performance of five cultivars on estimated environmental value. Each color represents a different cultivar. Lines are fitted values of genotype by environment combination. The horizontal axis displays the estimated environmental effects and the vertical axis represents the yield performance.

**Table 6.** Analysis of variance of additive main effects and multiplicative interaction (AMMI) for yield on the genotype and environment interactions.

| Source | DF | SS | MS | F-Value | *p*-Value | Variation Explained (%) |
|---|---|---|---|---|---|---|
| Environment € | 12 | 95,555,434 | 7,962,953 | 26.57969 | *** | |
| R€(E) | 39 | 19,263,173 | 493,927.5 | 1.64869 | 0.015 | |
| Genotype (G) | 4 | 16,411,368 | 4,102,842 | 13.69496 | *** | |
| G × E | 48 | 29,139,325 | 607,069.3 | 2.026348 | *** | |
| PC1 | 15 | 12,765,739 | 851,049.3 | 2.84 | *** | 46.9 |
| PC2 | 13 | 6,723,011 | 517,154.7 | 1.73 | * | 24.7 |
| PC3 | 11 | 5,845,645 | 531,422.3 | 1.77 | | 21.5 |
| PC4 | 9 | 1,885,576 | 209,508.5 | 0.7 | | 6.9 |
| Residuals | 182 | 54,524,984 | 299,587.8 | | | |
| Total | 333 | | | | | |

PC1, PC2, PC3, and PC4 are the principal component axes 1, 2, 3, and 4, respectively. Significant at $p \leq 0.001$ (***); and $p \leq 0.05$ (*).

The G × E was highly significant and partitioned into four interaction principal component analysis axes (IPCA). The PC1 and PC2 scores were significant and explained 46.9% and 22.27% of variability relating to G × E, respectively. Biplots (Figures 8 and 9) [49] provide a graphical representation of genotypes and environment from AMMI analyses and reveal three mega-environments. The AMMI1 biplot (Figure 8) includes mean pod yield and PC1, whereas AMMI2 (Figure 9) includes PC1 vs. PC2.

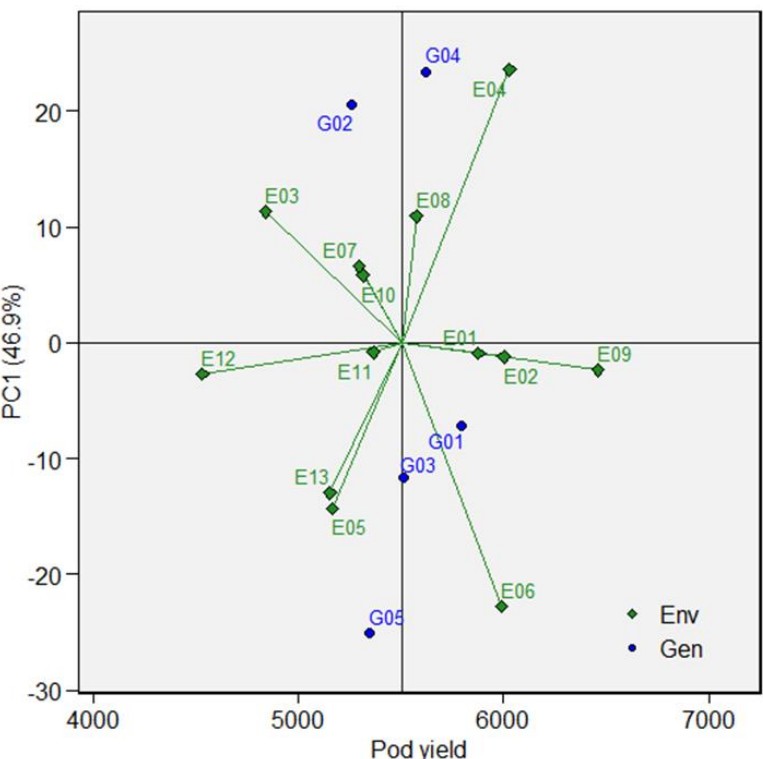

**Figure 8.** AMMI1 biplot (mean vs. PC1) for pod yield (kg/ha) with five peanut cultivars (G01—Bailey, G02—FloRun107, G03—Sullivan, G04—TUFRunner297, G05—Wynne) in 13 environments (E01—Cap17Rf, E02—Cap17RfIR, E03—Dwd16Rf, E04—Dwd16RfIR, E05—Rkm17Rf, E06—Rkm17RfIR, E07—Rkm18Rf, E08—Rkm18RfIR, E09—Suf17Rf, E10—Suf18Rf, E11—Suf18RfIR, E12—Suf20Rf).

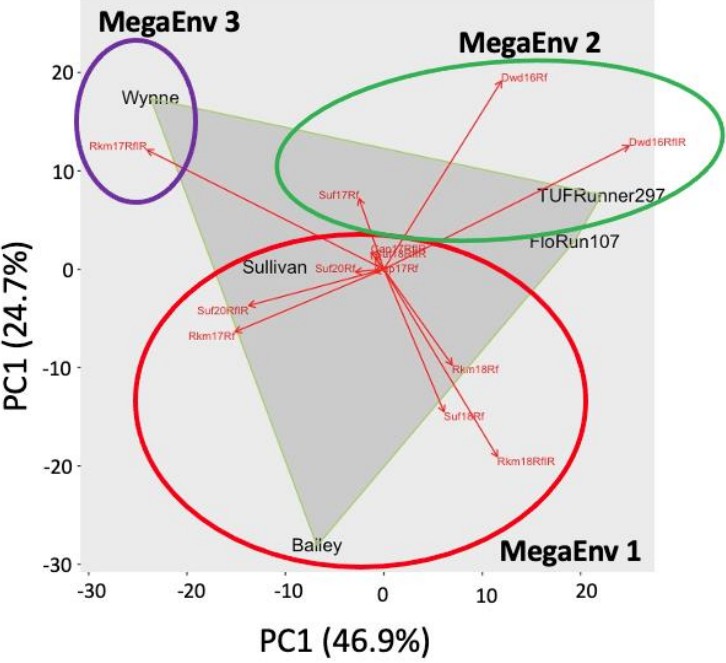

**Figure 9.** AMMI2 biplot (PC1 vs. PC2) for pod yield (kg/ha) with five cultivars (G01—Bailey, G02—FloRun107, G03—Sullivan, G04—TUFRunner297, G05—Wynne) in 13 environments (E01—Cap17Rf, E02—Cap17RfIR, E03—Dwd16Rf, E04—Dwd16RfIR, E05—Rkm17Rf, E06—Rkm17RfIR, E07—Rkm18Rf, E08—Rkm18RfIR, E09—Suf17Rf, E10—Suf18Rf, E11—Suf18RfIR, E12—Suf20Rf) showing mega-environments and their respective high yielding cultivars.

According to Figure 8, genotypes that were stable and, therefore, can be grown reliably in multiple locations, were G01 and G03, corresponding to Bailey and Sullivan. TUFRunner 297 (G04) was productive in terms of yield for most of the environments, whereas Wynne showed specific adaptability to environment E06 (Rkm17RfIR). TUFRunner 297 exhibited a similar response in E04 (Dwd16RfIR) (Figure 8). An AMMI 2 biplot was used for generating mega-environments. In the AMMI biplot, there were three mega-environments including Megaenv1 as Cap17Rf, Cap17RfIR, Rkm17Rf, Rkm18Rf, Rkm18RfIR, Suf18Rf, Suf18RfIR, Suf20Rf, and Suf20RfIR, with highest yielding cultivar Bailey; Megaenv2 including Dwd16Rf, Dwd16RfIR, and Suf17Rf with TUFRunner297; and Megaenv3 with Rkm17RfIR and Wynne as the best yielding cultivar (Figure 9).

## 4. Conclusions

The objective of this study was to evaluate five Virginia- and runner-peanut cultivars for pod yield stability by analyzing the G × E interaction over four years across 13 environments. Pod yield and grading factors were determined, and mean pod yield was used to determine yield stability using different stability models, including Linn and Binn's, Shukla, Wricke's, Finlay and Wilkinson, and AMMI. Sullivan and Bailey displayed higher adaptability and stability across multiple stability indices. Therefore, they could be recommended for reliable production across multiple environments in the VC region. Wynne and TUFRunner 297 presented high mean productivity. However, they were unstable and had specific adaptations to limited environmental conditions and could be recommended for specific locations. Environment E12 (Suf20Rf) gave the lowest mean yield and can be considered as unfavorable for peanut production, whereas environment E09 (Suf17Rf) had the highest mean yield and was indicated as the most favorable environment for peanut production in this study. Both environments are in Suffolk, VA, E12 being a year (2020) with prolonged drought from end-June to early August, and E09 a year (2017) with constant precipitation during the growing season and warm temperatures early in the season. The year 2017 was a successful year for peanut production in Virginia, when the average State yield was the record 5544 kg ha$^{-1}$. The SMK and TK for runners was higher than Virginia type. However, it changed by water regime in both market types. In 2020, SMK and TK under rainfed conditions reduced gradually in runners as compared to Virginia (Sullivan and Wynne), where the reduction was substantial. A slight increase in SS for both market types was noticed in the irrigated regime. This study showed that the peanut market types had broad and specific responses to the environments under investigation, and the decision to plant Virginia or runner-type needs to be based on the specific field history.

In conclusion, the main outcome of this work was to provide growers with information on the commercial cultivars' suitability that could allow them to make data-based decisions for planting. Nonetheless, this analysis provided a great exercise on the use of stability analyses for peanuts, that can be applied to other sets of cultivars, breeding populations, germplasm collections, and/or RIL populations to help breeders develop more stable cultivars across different environments in the VC region.

**Supplementary Materials:** The following supporting information can be downloaded at: https: //www.mdpi.com/article/10.3390/agronomy12123206/s1, The raw data for yield, SMK, SS, ELK, and TK can be found on Table S1.

**Author Contributions:** M.B., D.C.H. and N.K. conceptualized the project; N.K. mainly realized the hypothesis and objective development, with valuable suggestions and comments from M.B., D.C.H. and J.C.D.; N.K. collected and analyzed the data and wrote the manuscript; M.B., D.C.H. and J.C.D. provided a thorough review of the data analysis and manuscript. All authors have read and agreed to the published version of the manuscript.

**Funding:** This research was funded by USDA NIFA AWARD # 2016-08666.

**Institutional Review Board Statement:** Not applicable.

**Informed Consent Statement:** Not applicable.

**Data Availability Statement:** Not applicable.

**Acknowledgments:** The authors would like to thank field technicians Doug Redd, Frank Bryant, Cherry Fitz, and Zoe Dunlow for their help in conducting field operations and management of the peanut plots.

**Conflicts of Interest:** The authors declare no conflict of interest.

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
