# Peer review of "Multilocation Evaluation of Virginia and Runner -Type Peanut Cultivars for Yield and Grade in Virginia–Carolina Region"

_agronomy, doi:10.3390/agronomy12123206_

Round 1

Reviewer 1 Report

General:

This paper is nicely drafted and easy to read.

Poor formatting (Landscape orientation of all pages) and without line number it is difficult to highlight line number.

Abstract:

Line 1: tropical regions

Line is not mentioned so tough to highlight: Varied in yield ( ) for ….cultivars

Introduction: [13-15] Please highlight what do you mean by increases the GXE interaction?

Page 7: Please arrange formula in last line

Discussion

I would suggest the authors to have a more supported discussion with references considering the main point: The limitations of method and considerations when to apply the studied methodology and then the potential next steps or further investigation to address these limitations.

References: Please double check the style of references and missing one

Author Response

Please see the atttachment.

Reviewer 2 Report

This study identified cultivars that maintain high yield across a wide range of environmental conditions. They evaluated five commercially available virginia and runner-type peanut cultivars for pod yield stability using multilocation trials over four years across 13 environments and used different stability approaches study genotype (G), environment (E), and their interaction (G × E) on pod yield. At the same time, the grade factors of different conditions were determined. There were 4 inter-annual experiments, and the amount of work invested in the study was large. The article is well written, but there are some problems, which must be solved before it is considered for publication. the following problems should be well-addressed.

1.     The abstract emphasizes too much the importance, significance, and methods of the research, but the results of the paper are seriously insufficient, so that the readers cannot understand the final results well. The study results of grade factors are the important part of the article, please summarize them in the conclusion.

2.     The experimental design of this paper was rather confusing. Although the author tried to evaluate the stable yield characteristics of different peanut varieties by evaluating multi-year fractions, some varieties lacked year repetition and some varieties lacked irrigation treatment, which made the experiment very incomplete. In addition, the irrigation time and amount are not specified

3.     And Cultivars were planted in two-row plots of 10.6 m long × 0.9 m wide, this design makes the marginal effect very significant and hardly reflects the true characteristics of the variety. Moreover, the use of combine harvester for harvest and yield measurement in this plot experiment has a great impact on the yield.

4.     How is gross income calculated and obtained in Figure 2? The author does not explain clearly, More economic factors (irrigation costs) and calculation methods should be mentioned.

5.     Results and Discussion: “whereas in 2020 the month of July had a rainfall deficit, which affected peg and pod development by prolonging the vegetative phase of peanuts (Table 3).” This statement is not supported by the data. Please list the relevant research data about the peg and pod or delete it.

6.     Irrigation and non-irrigation were included in the design of the experiment, but only the environment and genotype were considered in the analysis of the results. Anova of treatment, environment and genotype should be added.

7.     Many of the figures in the article show the study results of grade factors, which should be mentioned in the abstract.

8.     Introduction : “The goal of this study was to identify cultivars that maintain high yield across a wide range of environmental conditions.” The goal and conclusions in this article both discuss high yield of cultivars in different conditions. But the study results of grade factors are listed in the article. What is the significance of studying grade factors? What is the relationship between high yield and grade factors? Please discuss it.

9.     Conclusions: “Environment E12 (Suf20Rf) gave the lowest mean yield and can be considered as unfavorable for peanut production, whereas environment E09 (Suf17Rf) had the highest mean yield and was indicated as the most favorable environment for peanut production in this study.” Soil moisture as an environmental condition that can be changed, irrigation is set in this study. However, the highest and lowest yields were not in the irrigation treatment nor is it mentioned in the conclusion. Would it be better to explain the reasons for not appearing in the irrigation treatment and the effect of irrigation on yield.

10.   Minor things need to be paid attention: Figure 2-6: “Dwd” in the figure is inconsistent with the notes “Din”, this is not good for understanding.

Reviewer 3 Report

Figures 2 and 3 looks very similar, try to combine results of both pod yield and gross income into 1 figure.

Include raw data for yield, SMK (%), SS, Total Kernel (%) in their respected results section as supplemental tables.

For environments Suf20Rf and Suf20RfIR, it would be interesting to compare yield loss percentages of the peanut varieties and observe relative yield, SMK (%), SS, Total Kernel (%) rankings to observe relative drought tolerance levels.  Calculate potential economic loss between irrigated/rainfall and between different peanut varieties.

Since this study only covers a few peanut lines, how would this result be applied to future peanut lines (for example peanut germplasm collection or new RILs) to identify more environmentally stable lines?  Discuss

Round 2

Reviewer 2 Report

Thank you for your response. These revisions to the manuscript basically responded my questions. We hope that the author will continue to improve and get better results in the next experiment.

Author Response

Thank you so much. I really appreciate your valuable insights and suggestions that improved the manuscript. This is my first manuscript and I will keep improving in my future writing as per your suggested comments.